# Nanotoxicity of 2D Molybdenum Disulfide, MoS_2_, Nanosheets on Beneficial Soil Bacteria, *Bacillus cereus* and *Pseudomonas aeruginosa*

**DOI:** 10.3390/nano11061453

**Published:** 2021-05-31

**Authors:** Michael Bae, Jun Kyun Oh, Shuhao Liu, Nirup Nagabandi, Yagmur Yegin, William DeFlorio, Luis Cisneros-Zevallos, Ethan M. A. Scholar

**Affiliations:** 1Artie McFerrin Department of Chemical Engineering, Texas A&M University, College Station, TX 77843, USA; bsy7790@tamu.edu (M.B.); liushuhao1993@tamu.edu (S.L.); nirup.nagabandi@essentium.com (N.N.); yagmur-ravli@tamu.edu (Y.Y.); wdeflorio@tamu.edu (W.D.); 2Department of Polymer Science and Engineering, Dankook University, 152 Jukjeon-ro, Suji-gu, Yongin-si 16890, Gyeonggi-do, Korea; junkyunoh@dankook.ac.kr; 3Department of Nutrition and Food Science, Texas A&M University, College Station, TX 77843, USA; lcisnero@tamu.edu; 4Department of Horticultural Science, Texas A&M University, College Station, TX 77843, USA; 5Department of Materials Science and Engineering, Texas A&M University, College Station, TX 77843, USA

**Keywords:** MoS_2_ nanomaterials, 2D nanosheets, nanotoxicity, soil bacteria

## Abstract

Concerns arising from accidental and occasional releases of novel industrial nanomaterials to the environment and waterbodies are rapidly increasing as the production and utilization levels of nanomaterials increase every day. In particular, two-dimensional nanosheets are one of the most significant emerging classes of nanomaterials used or considered for use in numerous applications and devices. This study deals with the interactions between 2D molybdenum disulfide (MoS_2_) nanosheets and beneficial soil bacteria. It was found that the log-reduction in the survival of Gram-positive *Bacillus cereus* was 2.8 (99.83%) and 4.9 (99.9988%) upon exposure to 16.0 mg/mL bulk MoS_2_ (macroscale) and 2D MoS_2_ nanosheets (nanoscale), respectively. For the case of Gram-negative *Pseudomonas aeruginosa*, the log-reduction values in bacterial survival were 1.9 (98.60%) and 5.4 (99.9996%) for the same concentration of bulk MoS_2_ and MoS_2_ nanosheets, respectively. Based on these findings, it is important to consider the potential toxicity of MoS_2_ nanosheets on beneficial soil bacteria responsible for nitrate reduction and nitrogen fixation, soil formation, decomposition of dead and decayed natural materials, and transformation of toxic compounds into nontoxic compounds to adequately assess the environmental impact of 2D nanosheets and nanomaterials.

## 1. Introduction

Prior research has indicated that engineered nanomaterials, such as quantum dots, nanoparticles, nanowires, nanorods, and nanosheets, can be released to the environment contingently during their life cycles (product use, disposal, and weathering) [1]. The increasing use of engineered nanomaterials has led to an increasing concern on their possible build-up in the environment, and sequentially in the food supply [2,3]. Although various unique properties of nanomaterials have made them attractive in numerous applications, some of these properties, such as enhanced transport, increased bioavailability, enlarged surface area, and greater surface reactivity, can adversely impact living organisms and microorganisms [4]. For example, Dimkpa et al. [5] investigated the effect of CuO (<50 nm) and ZnO (<100 nm) NPs on wheat (*Triticum aestivum*) grown in a solid matrix (sand). They reported oxidative stress in NP-treated plants, which was proven by increased lipid peroxidation and oxidized glutathione in roots and decreased chlorophyll content in shoots, and higher peroxidase and catalase activities in roots. Xin et al. [6] reported that silver nanoparticles can modify the expression profiles of neural development-related genes (*gfap*, *huC* and *ngn1*), metal-sensitive metallothioneins, and ABCC genes in exposed zebrafish embryos. However, the majority of previous efforts with environmental implications of nanomaterials on living organisms have primarily focused on metal and metal oxide nanomaterials, as well as carbon-based nanomaterials such as fullerenes, graphene, and carbon nanotubes (CNTs) [7,8,9,10,11].

Recently, among various types of nanomaterials, 2D nanosheets such as graphene, hexagonal BN, MoS_2_, WS_2_, MgO_2_, MXenes, owing to their ultrahigh aspect ratio and unusual optical, electronic, thermal, and mechanical properties, have gained particular attention from scientists and engineers [12,13,14,15,16]. Rapidly emerging applications relying on 2D nanosheets include electrodes for energy storage devices [17], lubricants and friction reducers [18], thermal management materials, [15], sensors [19], membranes for gas separation [20], water purification systems [21], composite structural materials [22], and oil cleanup systems [23]. With an increasing number of applications and utilization, it is essential to study and understand the interactions of 2D nanosheets with living organisms and environmental surfaces in order to assess the impact of such nanomaterials on the environment. Accordingly, many studies have recently investigated the interactions of 2D nanosheets (primarily graphene) and living organisms such as plants, animals, algae, and bacteria [24,25,26].

Molybdenum disulfide (MoS_2_) nanosheets, which are relatively new types of 2D nanomaterials, have received increased attention after the peak graphene and hexagonal boron nitride (h-BN) era. The bulk form of MoS_2_ exhibits randomly stratified poly-structures because of its hexagonal monolayers bound to each other via Van der Waals forces [12]. The hexagonal monolayer of MoS_2_ has a transition metal in the middle, sandwiched by two layers of chalcogenide atoms with a stable covalent bonding [27]. The bulk MoS_2_ has a stacked multilayer form in nature, which can readily be exfoliated into fewer layers with the exertion of mechanical force or agitation [28,29]. MoS_2_ nanosheets have the ability to act as a catalyst for generating hydrogen [30,31], which can be applied to hydro-desulfurization processes in industry [32]. The enhanced bandgap of MoS_2_ nanosheets (1.8 eV) has been associated in photoelectric reactions [33,34]. The capability of disinfecting water by harvesting the whole spectrum of visible light from the sun has also been noted as one of the applications with MoS_2_ nanosheets [35]. The ability to prevent bacterial biofilms with MoS_2_ particles (90 nm and 2–6 µm) and surfaces has recently been reported [36].

Soil bacteria are a fundamental and integral part of soil ecological systems. They control carbon dynamics, nutrient cycles, nitrification and denitrification, the early stages of decomposition of organic materials, and plant productivity [37,38]. In addition, they take up and transform toxic compounds into nontoxic compounds via immobilization [39,40]. Therefore, plausibly, recent studies have focused on the impact of engineered nanomaterials on soil bacteria. Disruption of cell walls, hydrophobic interactions with cell membranes, and the dissolution/release of metals have been proposed as mechanisms for the nanotoxicity of engineered nanomaterials such as CuO, ZnO, MgO, CeO_2_ nanoparticles (NPs) on soil bacteria [41,42,43]. Whiteside et al. [44] and Moll et al. [45] found that CdSe/ZnO quantum dots and CeO_2_, TiO_2_ NPs had no effect on nitrogen-fixing bacteria nodulation, while Sillen et al. [43] found significant differences in bacterial community carbon use and lowered enzymatic activity upon exposure to ZnO, CeO_2_ NPs. However, studies focusing on how MoS_2_ nanosheets interact with soil bacteria and what their potential toxic effects are on soil bacteria have not been reported, to the best of our knowledge.

In this work, we investigated the growth dynamics of Gram-positive *Bacillus cereus* and Gram-negative *Pseudomonas aeruginosa* under the influence of bulk MoS_2_ and 2D MoS_2_ nanosheets as a function of concentration and exposure time. *B. cereus* was chosen as the model soil bacteria because they are involved in nitrification processes, phosphate transport, protection of the rhizosphere from fungal diseases, and heavy metal remediation [46,47,48]. *P. aeruginosa* was selected as another model soil bacteria in this study because they play a central role in nitrate reduction and the decomposition of toxic compounds in soil [49,50]. Bacterial growth behavior in the presence and absence of bulk MoS_2_ and 2D MoS_2_ nanosheets were investigated using the agar plating assay. Then, by analyzing concentration-dependent bacterial survival data, the median effective concentration (EC50), corresponding to the concentration of bulk and nanoexfoliated MoS_2_ which induces a response halfway between the baseline and maximum after 8 h of exposure time, was calculated for each microorganism from the dose–response curve. Scanning electron microscopy was used to obtain complementary information on how bulk and nanoexfoliated MoS_2_ influences bacterial morphology. Zeta potential measurements were conducted to gain insights into the nature of intermolecular interactions between MoS_2_ and soil bacteria.

## 2. Materials and Methods

### 2.1. Preparation of MoS_2_ Nanosheets

Molybdenum (IV) disulfide, 99% (metal basis) powder was purchased from Alfa Aesar (CAS No. 41827, Haverhill, MA, USA) (Appendix A). After MoS_2_ was added to deionized water (DI) water at a concentration of 16 mg/mL, high-intensity ultrasonication was utilized to exfoliate bulk materials into nanosheets via a probe sonicator (SJIA-2000W, Ningbo Haishu Sklon Electronics Instruments Co., Zhejiang, China). The exfoliation was achieved at a sonication power of 2000 W and frequency of 19.5–20.5 Hz with one hour of ultrasonication (3 s on and 1 s off cycles). All of these processes were carried out in an ice bath to eliminate the possibility of temperature-induced oxidation of MoS_2_ (The exfoliated MoS_2_ in water information is uploaded in Appendix A). On the other hand, to prepare the bulk controls, the bulk suspension was centrifuged at 4000 rpm for 15 min (AccuSpin 400, Thermo Fisher Scientific, Waltham, MA, USA) to separate nanoscale materials from the supernatant layer, and the same volume of water was compensated.

### 2.2. Bacterial Cultures and Plate Counting

Two soil bacteria, Gram-positive *Bacillus cereus* (ATCC 14579) and Gram-negative *Pseudomonas aeruginosa* (ATCC 9027), were used in this study. Experimental cultures of these were transferred by using the tip of an inoculating loop (CAS No. 12000-812, VWR, 10 μL, sterile), from tryptic soy agar (TSA; Becton, Dickinson and Co., Franklin Lakes, NJ, USA) slant with grown colonies to the culturing centrifuge tube which contained 9 mL of tryptic soy broth (TSB; Becton, Dickinson and Co., Franklin Lakes, NJ, USA). The tubes of these bacteria were incubated aerobically at 37 °C for 24 h without shaking. Next, an inoculating loop transferred 10 μL from the 9 mL of tryptic soy broth with bacteria to a fresh 9 mL of tryptic soy broth medium. This process repeated up to three times. Second and third transfers of the culture were cleaned with sterilized DI water for twice after performing centrifugation for 15 min each with 4000 rpm to remove the supernatant part of tryptic soy broth and replacing it with sterilized DI water twice, then refilling it with 1 g/L peptone (Becton, Dickinson and Co., Franklin Lakes, NJ, USA) water in the end. The final growth in the culture media after plate counting was 6.48 ± 0.15 log_10_ CFU/mL for *B. cereus* and 8.49 ± 0.08 log_10_ CFU/mL for *P. aeruginosa* by stirring them with a magnetic bar for 24 h at room temperature, transferring 1 mL to a Petri dish, mixing them with TSA, and incubating them again for 24 h at 37 °C and counting the number of the colonies on the agar (Appendix A).

### 2.3. Characterization of Soil Bacteria

For characterization, (100) silicon wafers were cut into 10 mm × 10 mm pieces and then polished by piranha solution with a 3:1 mixture of sulfuric acid and 30% hydrogen peroxide for 1 h, washed with ultrapurified water, and left to dry at 23 °C. The cultured bacteria that were diluted in sterilized DI water at a volumetric ratio of 1000:1 were drop-cast with 1~3 drops using a micropipette (100 μL) to cover the silicon wafer surface, exposed to a trace amount of acrolein, dried for one day inside the biological safety cabinet at room temperature, and imaged with a scanning electron microscope (JSM-7500F; Jeol USA, Peabody, MA, USA) for visualizing the cell surface. Before SEM imaging, 5 nm palladium and platinum (Pd/Pt) alloys were deposited on the surfaces to ensure the electrical conductivity for SEM measurements and to immobilize adhered bacteria cells.

### 2.4. Characterization of MoS_2_

The morphology and size characteristics of bulk MoS_2_ and MoS_2_ nanosheets were determined using scanning electron microscopy (SEM) and atomic force microscopy (AFM). In both cases, after MoS_2_ nanosheet stock solution was diluted 100-fold in DI water, a droplet of suspension was placed on a (100) silicon water, followed by 24 h drying at room temperature. The samples were characterized with an atomic force microscope (AFM, Bruker Dimension Icon, Billerica, MA, USA) using the tapping mode at room temperature in standard air atmosphere. The measurements were performed with a silicon tip (OMCL-AC200TS-R3, Olympus, Center Valley, PA, USA) which had a radius of curvature of 7 nm, a spring constant of 9 N/m, and a resonant frequency of 150 kHz, at a scan rate of 0.5 Hz. SEM measurements of MoS_2_ samples were carried out similarly to the SEM characterization of bacteria, although no conductive layer was used for the MoS_2_ samples.

The pH of MoS_2_ suspension in water assisted by magnetic bar stirring was measured with a pH meter (S20 SevenEasy pH, Mettler Toledo, Columbus, OH, USA). The pH measurements were carried out as a function of suspension age (initial, 2 h, 4 h, 8 h, 12 h, and 24 h) and concentration (1.6 mg/mL, 4.0 mg/mL, 8.0 mg/mL, 16.0 mg/mL) for both bulk MoS_2_ and nanoexfoliated MoS_2_. All measurements were repeated at least three different times from different batches of samples to enable statistical analysis to be performed.

### 2.5. Zeta Potential Measurements

The zeta potential of the samples, which is the potential at the slipping/shear plane of colloid particle movement assisted by electric field energy [51], was measured with a dynamic light scattering (DLS) instrument (Malvern Instruments, Ltd., Malvern, UK) at 25 °C. The zeta potential was calculated from the Helmholtz–Smoluchowski equation using the electrophoretic mobility, dynamic viscosity of the continuous phase, and dielectric permittivity factor at the liquid and vacuum of the continuous phase [51,52]. All colloidal entities (bulk MoS_2_, exfoliated MoS_2_, *Bacillus cereus*, and *Pseudomonas aeruginosa*) were diluted to a lower concentration of 0.1 vol% to 1 vol% for the zeta potential measurements.

### 2.6. Analysis of MoS_2_ Toxicity and Dose–Response Analysis

To observe overall survivability trends, two different soil bacteria were inoculated with bulk and exfoliated MoS_2_ at five different concentrations (0, 1.6, 4.0, 8.0, and 16.0 mg/mL) and six different time points (0 h, 2 h, 4 h, 8 h, 12 h, and 24 h). A transfer of 1 mL was taken from samples which were incubated with the mixture of peptone water and MoS_2_ suspension with a stirring magnetic bar for a pre-defined period of exposure time. Then, the 1 mL mixture was filled with warm TSA (40 °C) in a new Petri dish and gently shaken. Upon solidification for 10 min, the samples were placed in an aerobic incubation chamber at 37 °C for 24 h. Experiments were conducted at least in triplicates (up to seven repeats) for each concentration and time condition. The number of grown colonies after 24 h was counted from the TSA Petri dish by multiplying its dilution rate. The bacterial survival numbers, *N*, were normalized based on the initial colony number, *N*_0_, for each treatment (Equation (1)).
(1)$$N_{normalized\;survival}=\frac{N}{N_{0}}$$

Dose–response curves were analyzed using a Sigmoidal model, as shown in Equation (2), with the aid of Origin software (Origin Pro 8, OriginLab Corp., Northampton, MA, USA).
(2)$$y=min+\frac{max-min}{\left(1+10^{LogEC50-X}\right)}$$
where *X* is the logarithm of MoS_2_ concentration, *min* is the bacterial survival number at the bottom plateau, and *max* is the bacterial survival number at the top plateau.

### 2.7. Statistical Analysis

To obtain the average and standard deviations, the statistical package in Analysis ToolPak-Excel (Microsoft Corp., Redmond, WA, USA) was used. For comparing the statistical differences in the survivability of *B. cereus* and *P. aeruginosa*, two-way analysis of variance (ANOVA) with Tukey’s post hoc test was utilized to determine the statistical similarity of the data sets with *p*-values.

## 3. Results and Discussion

### 3.1. Characterization of Soil Bacteria

In many physicochemically interacting systems, the interplay among relevant characteristic length scales such as particle size, particle thickness, bacterial diameter, and bacterial size often controls the dynamics of interactions. Accordingly, we first investigated the morphological characteristics of soil bacteria. Figure 1a,b shows SEM micrographs of *B. cereus* and *P. aeruginosa* in the absence of any exposure to MoS_2_. The average size of *B. cereus* was slightly larger than that of *P. aeruginosa* for both length and the diameter described in Table 1. The average length and diameter of *B. cereus* was 2.86 ± 0.83 μm and 0.79 ± 0.10 μm, respectively; the average length and diameter of *P. aeruginosa* was 2.31 ± 0.41 μm and 0.63 ± 0.18 μm. Furthermore, extracellular polymeric substances (EPS) excreted by bacteria, which contain polysaccharides, proteins, nucleic acids, lipids, and other macromolecules to assist the adaptation of the cells by making them attach and aggregate on the surfaces [53,54], could also be seen from these SEM micrographs.

Zeta potential value is an important parameter that controls the interfacial behavior of bacteria such as the colloidal stability and adhesion [55,56]. Figure 1c indicates the zeta potential of *B. cereus* and *P. aeruginosa* in DI water. *B. cereus* has a zeta-potential of −33.3 ± 1.1 mV, whereas *P. aeruginosa* has a zeta potential of −44.3 ± 1.2 mV. These values are sufficiently large that colloidal aggregation of these bacteria is unlikely to occur. The differences in the zeta potential can be attributed to the differences in Gram-positive and Gram-negative bacterial walls. Gram-positive bacteria are negatively charged due to the presence of teichoic acid containing glycerol or ribitol phosphates which can contribute to the antibiotic susceptibility of bacteria [57,58]. On the other hand, Gram-negative bacteria are negatively charged because of the presence of lipopolysaccharides, which provide them with their adhesive ability for survival and stabilizing their outer membrane to protect the inner structure [59,60].

### 3.2. Characterization of MoS_2_

Comparison of size characteristics of MoS_2_ with bacteria is important to gain insights into the relative mobility of MoS_2_ and bacteria in aqueous media and the surface potential of MoS_2_ to cover bacteria surfaces. Figure 2 demonstrates micrographs of bulk MoS_2_ and MoS_2_ nanosheets as well as their zeta potentials. Image analysis over multiple samples (summarized in Table 2) revealed that bulk MoS_2_ has a mean diameter of 12.0 ± 7.6 μm (geometric mean) and a thickness of 520 ± 364 nm. On the other hand, the diameter and thickness of MoS_2_ nanosheets were 0.88 ± 0.81 μm and 3.1 ± 0.7 nm, respectively. These size characteristics indicated that the ultrasonication process not only separated the layers, but also broke the layers into smaller fragments in the planar direction. In the existing literature, nanomaterials with a thickness of less than 1–10 nm, as in the case of exfoliated MoS_2_ in this study, are often categorized as 2D nanosheets [31,61]. Given that the thickness of each MoS_2_ layer is reported to be 0.65 nm [62], a thickness of 2 to 4 nm corresponds to three to six layers.

As can be seen from Figure 2c, zeta potential values of −18.4 ± 1.5 mV and −25.4 ± 0.2 mV were observed for bulk MoS_2_ and exfoliated MoS_2_, respectively. Accordingly, the electrostatic interactions between MoS_2_ and bacteria are repulsive. For the case of interactions between MoS_2_ particles and bacteria in water, even if the overall interaction electrical charge between bacteria and MoS_2_ is repulsive, there is a Boltzmann probability of MoS_2_ adhesion on bacteria (or bacterial adhesion on MoS_2_) governed by the magnitude of the activation energy in terms of kT [63,64]. Furthermore, prior studies have indicated that nanosheets can orient themselves perpendicularly to the surfaces during the approach to significantly reduce the magnitude of repulsion [14]. For instance, the deposition of negatively charged graphene oxide on self-assembled monolayers of 6-aminohexy-aminopropyltrimethoxysilane, which possess a negative zeta potential above pH ~6, was confirmed via AFM studies [14,65,66].

### 3.3. Survival of B. cereus and P. aeruginosa against MoS_2_ Exposure

As can be seen from Figure 3, the addition of MoS_2_ into *B. cereus* suspension resulted in a concentration-dependent reduction in bacterial survival. For the case of bulk MoS_2_ exposure, the survival data followed an initially rapidly decreasing trend, which gradually plateaued out after an exposure time of 8 h. The log reduction in survival upon 24 h exposure was ~0.3 (55% reduction) and ~2.6 (99.7% reduction) at a MoS_2_ concentration of 1.6 mg/mL and 16.0 mg/mL, respectively. For the case of exfoliated MoS_2_, similar trends were also observed, but the log-reductions in survival numbers were much larger: ~0.9 (86% reduction) at 1.6 mg/mL and ~4.9 (99.999% reduction) at 16.0 mg/mL.

Similar to the studies with Gram-positive *B. cereus*, the influence of MoS_2_ on the survival of Gram-negative *P. aeruginosa* was also investigated (Figure 4). At a concentration of 16.0 mg/mL, the log-reduction in bacterial survival was ~1.9 (98.6%) and ~5.5 (99.9997%) for bulk MoS_2_ and exfoliated MoS_2_, respectively. Overall, *P. aeruginosa* demonstrated a slightly higher survival rate than *B. cereus* against bulk MoS_2_, whereas exfoliated MoS_2_, above a concentration of 1.6 mg/mL, resulted in the lower survival of *P. aeruginosa* compared to *B. cereus*.

Based on the statistical analysis, the toxicity levels of bulk MoS_2_ and exfoliated MoS_2_ were found to be statistically different for both bacteria (*p* < 0.05, see the Appendix A for further details: Appendix A). In addition, the toxicity of exfoliated MoS_2_ was always higher than bulk MoS_2_ (see the Appendix A for further details, Appendix A).

The gradually plateauing trends observed in these cases can be attributed to the following possibilities. First, given that a predefined amount of MoS_2_ exists in the suspension, continuous adsorption/uptake of MoS_2_ on/in bacteria gradually reduces the MoS_2_ concentration in the suspension. As new bacteria grow, the effective concentration of MoS_2_ becomes less and less with time. This could explain the plateauing trend and trends that the survival increases at a sufficiently long time. Secondly, the presence of bacteria and excreted extracellular polymeric substances (EPS) can induce the aggregation of MoS_2_ and the encapsulation/coverage of MoS_2_ with the EPS layer, which can reduce the surface dissociation processes and effective solubilization (i.e., bioavailability). Similarly, chemical changes can also take place on MoS_2_ in the presence of EPS. These effects, in turn, can reduce the potency of MoS_2_ as a toxic agent to bacteria.

Based on the analysis of the data shown in Figure 3 and Figure 4, a dose–response curve was constructed for bulk MoS_2_ control and exfoliated MoS_2_ (Figure 5 and Appendix A). The response curve relied on 8 h data as the suspension seemed to be depleted/sedimented for longer durations. It was found that the median effective concentration (EC_50_) was 1.81 ± 0.41 mg/mL and 1.00 ± 0.43 mg/mL for the case of bulk MoS_2_ against *B. cereus* and *P. aeruginosa*, respectively. On the other hand, for the case of MoS_2_ nanosheets, EC_50_ values of 1.45 ± 0.19 mg/mL and 0.59 ± 0.16 mg/mL were obtained *B. cereus* and *P. aeruginosa*, respectively. By comparing the median minimum bactericidal concentrations (MBC_50_) for other antimicrobial agents from the literature, such as clindamycin (1.0 µg/mL), gentamicin (2.0 µg/mL), vancomycin (2.0 µg/mL) for *B. cereus* [67], and ceftazidime (128.0 µg/mL), tobramycin (16.0 µg/mL), meropenem (16.0 µg/mL), aztreonam (128.0 µg/mL), piperacillin (64.0 µg/mL) for *P. aeruginosa* [68], it can be stated that EC_50_ values of bulk and exfoliated MoS_2_ are one to three orders of magnitude higher, indicating the relatively weak bacterial toxicity of MoS_2_. However, at sufficiently high concentrations (>~1 mg/mL), bulk and nanoexfoliated MoS_2_ can moderately inhibit the growth of *B. cereus* and *P. aeruginosa*.

### 3.4. Mechanism of Interaction between MoS_2_ and Bacteria

To gain a mechanistic understanding of how MoS_2_ interacts with and inactivates bacteria, SEM studies were performed (Figure 6, Appendix A). It was found that bulk MoS_2_ acts as a geometrical obstacle that hinders the formation of bacterial microcolonies and confines bacteria. In the presence of bulk MoS_2_, no significant change in the morphology of bacteria was observed. On the other hand, exfoliated MoS_2_ induced wrinkles and rhytids on a bacterial wall. Furthermore, due to its smaller size and thinner nature, exfoliated MoS_2_ could better conform to the curvature of bacteria. The mean length of bacteria exposed to exfoliated MoS_2_ was smaller than that exposed to bulk MoS_2_ and no treatment.

The presence of oxidative stress, which can be induced by MoS_2_ via the formation of oxides and sulfate ions [69,70] on the bacteria, could be the reason for wrinkles and rhytids on the cell. The local oxidation and etching/erosion of the cell wall can cause mechanical instabilities where the internal osmotic pressure can locally push thinner regions outward (i.e., nano-/micro-bulging) while the intact regions of the cell wall may remain mostly unaltered. Kaur et al. [56] observed the fragmentation and damage of cell walls for MCF7 (breast cancer), U937 (leukemia), HaCaT (epithelium), and *Salmonella typhimurium* upon inoculation with MoS_2_ nanosheets at a concentration of 10–20 µg/mL. Pandit et al. [71] reported the antibacterial activity of quaternary amine-functionalized, chemically exfoliated MoS_2_ nanosheets against *Staphylococcus aureus* and *Pseudomonas aeruginosa*, whereas hydroxyl-functionalized MoS_2_ nanosheets showed no antibacterial activity. These findings suggest that ligands rather than MoS_2_ play a larger role in antibacterial activity. In addition, each bacterium can exhibit a different favorable environmental condition, such as mesophilic, thermophilic, acidophilic, and alkaliphilic conditions. Some bacteria can survive in oxidative stress conditions owing to their defensive systems [72]. *B. cereus* and *P. aeruginosa* are known to be mesophiles. However, *B. cereus* possesses gene clusters responsible for the arginine deiminase metabolic pathway, which is believed to play a pivotal role in resisting acidic conditions [73,74,75]. The range of pH allowing growth of *B*. *cereus* was reported to be pH 4.9 to 9.3 [76]. Based on acid treatment studies, Bushell et al. [77] reported that the growth rate of *P. aeruginosa* does not change much between pH 7 and 6, while noticeable reductions in bacterial growth are observed below pH 5.5, and almost no growth occurs at pH 5. Accordingly, we have also investigated the change in dispersion pH upon the addition of bulk and exfoliated MoS_2_.

It was found that the presence of bulk MoS_2_ reduced the dispersion pH to 6.3 and 4.3 at concentrations of 1.6 mg/mL and 16.0 mg/mL, respectively (Appendix A). In contrast, dispersion pH values of 5.4 and 3.5 were attained for the case of MoS_2_ nanosheets at concentrations of 1.6 mg/mL and 16.0 mg/mL, respectively, indicating a more acidifying potential of exfoliated MoS_2_. Two potential dissociating processes of MoS_2_, one involving a 1:10 MoS_2_:H_2_O ratio and another involving a 1:12 MoS_2_:H_2_O ratio, were described by Titley et al. [78] and Wagman et al. [79] with dissociation equilibrium constants of −117.89 and −131.74, as shown in Equations (3) and (4).
(3)$${\mathrm{MoS}}_{2}+10H_{2}O\;↔{\mathrm{MoO}}_{2}^{+}+2{\mathrm{SO}}_{4}^{2-}+20H^{+}+17e^{-};\;K=-117.89;$$
(4)$${\mathrm{MoS}}_{2}+12H_{2}O\;↔{\mathrm{MoO}}_{4}^{2-}+2{\mathrm{SO}}_{4}^{2-}+24H^{+}+18e^{-};\;K=-131.74;$$

In addition, sulfur atoms of MoS_2_ can be replaced by oxygen atoms via oxidation when kept in aqueous media for prolonged periods of time [69]. Liu et al. [80] reported that MoO_2_ has two kinds of proton-donating ligands, with pKa values of 4.7 (OH) and 10.6 (H_2_O) from MoO_2_(OH)_2_∙(H_2_O)_2_, which indicates that even after the transformation from MoS_2_, molybdenum oxides would persist in having an acidic nature. When plotted as growth trends, we can see that the higher concentrations of MoS_2_ give rise to environments that are unfavorable for the soil bacteria studied in this work. This means that apart from oxidative stress and blockage of the cell wall, acidity induced by the presence of MoS_2_ should also be considered in the context of toxicity of MoS_2_.

## 4. Conclusions

Two-dimensional nanosheets are important types of emerging nanomaterials receiving attention from various fields and applications. In this study, we investigated the toxicity of 2D MoS_2_ nanosheets (nanoscale) on soil bacteria *B. cereus* and *P. aeruginosa* and compared it with bulk MoS_2_ (macroscale) at various concentrations. It was found that 2D MoS_2_ nanosheets demonstrate a higher level of toxicity against these microorganisms than bulk MoS_2_. The 8 h EC_50_ value was 1.81 ± 0.41 mg/mL and 1.00 ± 0.43 mg/mL for bulk MoS_2_ against *B. cereus* and *P. aeruginosa*, respectively. In contrast, for exfoliated MoS_2_ nanosheets, EC_50_ values of 1.45 ± 0.19 mg/mL and 0.59 ± 0.16 mg/mL were obtained for *B. cereus* and *P. aeruginosa*, respectively. Three potential mechanisms of action have been identified as oxidative stress: the coverage and blockage of cell walls of nanosheets, the dissolution of MoS_2_, and the resulting acidification of the dispersion medium. Oxidative stress and acidification induced wrinkles and rhytids on bacterial walls. The blockage of cell walls, which was confirmed with SEM studies, can hinder nutrient transport and metabolic activities that occur on the cell wall. Compared to antimicrobial agents such as clindamycin, gentamicin, vancomycin, tobramycin, and piperacillin, EC_50_ values of bulk and exfoliated MoS_2_ are one to three orders of magnitude higher, indicating the relatively weak bacterial toxicity of MoS_2_. Overall, this study highlights the potential of MoS_2_ nanotoxicity on beneficial soil bacteria, which plays an essential role in nitrate reduction and nitrogen fixation, soil formation, the decomposition of dead and decayed natural materials, and the transformation of toxic compounds into nontoxic compounds.

## Figures and Tables

**Figure 1 nanomaterials-11-01453-f001:**
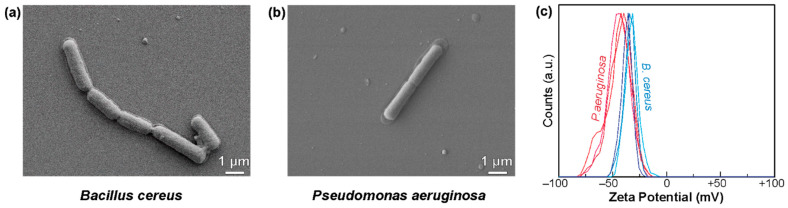
Scanning electron microscopy images of (**a**) *B. cereus* and (**b**) *P. aeruginosa*, and (**c**) zeta-potential of these microorganisms.

**Figure 2 nanomaterials-11-01453-f002:**
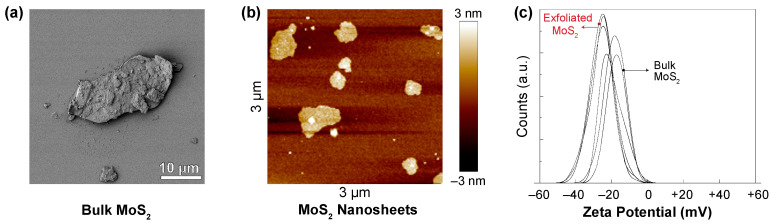
(**a**) Scanning electron microscopy image of bulk MoS_2_, (**b**) atomic force microscopy image of 2D MoS_2_ nanosheets, and (**c**) zeta potential of bulk and exfoliated MoS_2_.

**Figure 3 nanomaterials-11-01453-f003:**
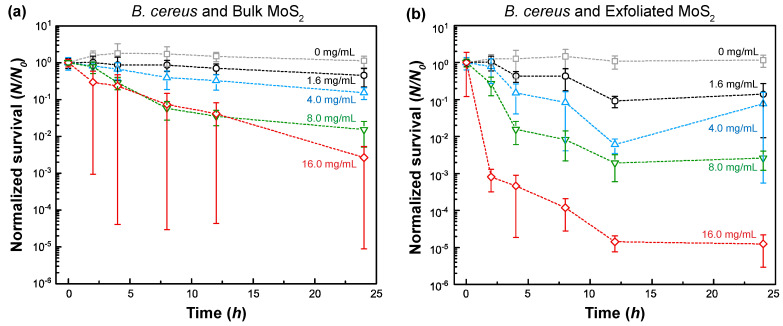
Normalized (with respect to initial concentration) survival of *B. cereus* in peptone water and MoS_2_ suspension inoculated with (**a**) bulk MoS_2_ and (**b**) exfoliated MoS_2_. The error bars represent the standard error of the mean.

**Figure 4 nanomaterials-11-01453-f004:**
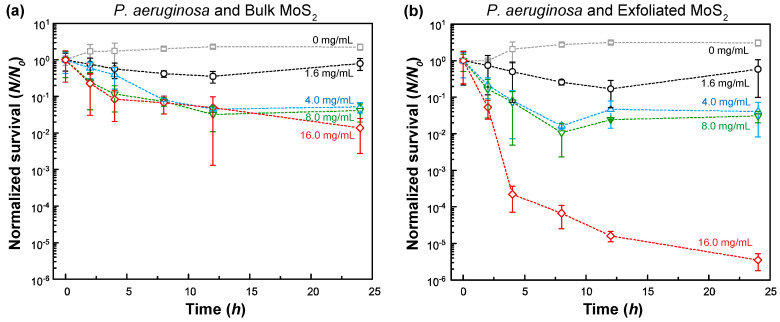
Normalized (with respect to initial concentration) survival of *P. aeruginosa* in peptone water and MoS_2_ suspension inoculated with (**a**) bulk MoS_2_ and (**b**) exfoliated MoS_2_. The error bars represent the standard error of the mean.

**Figure 5 nanomaterials-11-01453-f005:**
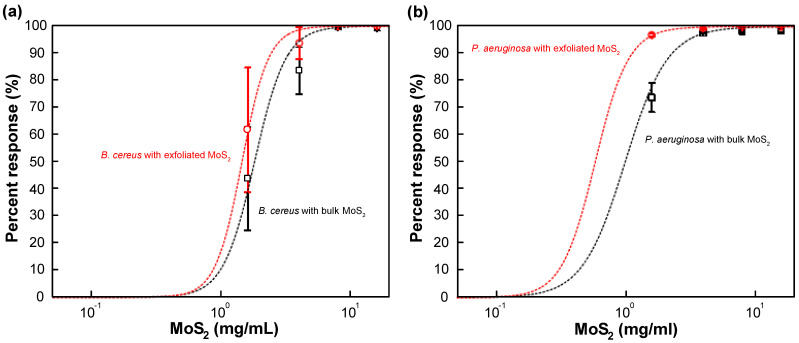
Dose–response curve of (**a**) *B. cereus* and (**b**) *P. aeruginosa* against bulk and exfoliated MoS_2_. The error bars represent the standard error of the mean.

**Figure 6 nanomaterials-11-01453-f006:**
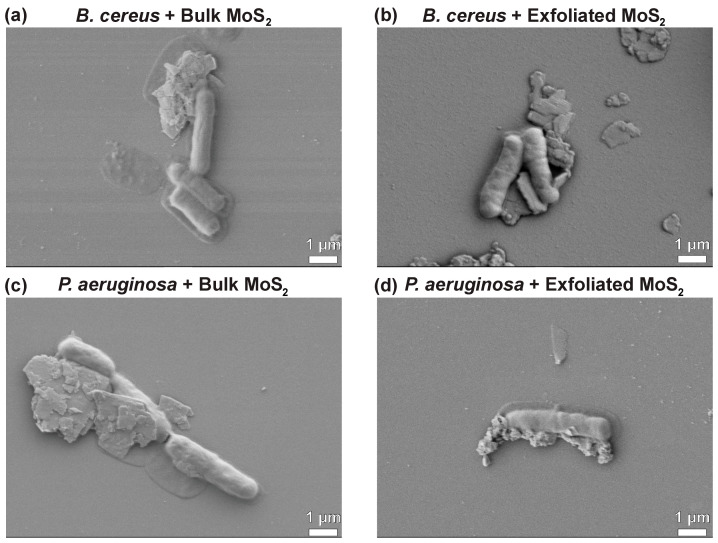
SEM micrographs showing interactions between soil bacteria and MoS_2_ particles at a concentration of 4.0 mg/mL for an inoculation time of 12 h. (**a**) *B. cereus* inoculated with bulk MoS_2_, (**b**) *B. cereus* inoculated with exfoliated MoS_2_, (**c**) *P. aeruginosa* inoculated with bulk MoS_2_, and (**d**) *P. aeruginosa* inoculated with exfoliated MoS_2_. Additional SEM images are shown in Appendix A, in the Appendix A.

**Table 1 nanomaterials-11-01453-t001:** The structural and surface characteristics of *B. cereus* and *P. aeruginosa* used in this study. ± values indicate the standard deviation.

Bacteria	*B. cereus*	*P. aeruginosa*
Type	Gram-positive	Gram-negative
Dimensions (μm)	W: 0.79 ± 0.10 L: 2.86 ± 0.83	W: 0.63 ± 0.18 L: 2.31 ± 0.41
Zeta potential (mV)	−33.3 ± 1.1	−44.3 ± 1.2

**Table 2 nanomaterials-11-01453-t002:** The structural and surface characteristics of bulk and exfoliated MoS_2_ used in this study.

MoS_2_	Bulk Form	Exfoliated Form
Planar diameter (μm)	12.0 ± 7.6	0.9 ± 0.8
Thickness (nm)	520 ± 364	3.1 ± 0.7
Zeta potential (mV)	−18.4 ± 1.5	−25.4 ± 0.2

± values indicate the standard deviation.

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
