# Peer review of "Nanotoxicity of 2D Molybdenum Disulfide, MoS2, Nanosheets on Beneficial Soil Bacteria, Bacillus cereus and Pseudomonas aeruginosa"

_nanomaterials, 2021, doi:10.3390/nano11061453_

Round 1
Reviewer 1 Report
- Review of the manuscript "Nanotoxicity of 2-D Molybdenum Disulfide, MoS2 Nanosheets on Beneficial Soil Bacteria, Bacillus cereus and Pseudomonas aeruginosa" by M. Bae et al.
The work deals with the bactericidal properties of molybdenum disulfide, and in particular studies the toxic activity of MoS2 (bulk vs exfoliated nanosheet samples) toward two bacteria, namely Bacillus cereus and Pseudomonas aeruginosa. These bacterial have been selected as object of study due to their beneficial functionalities in soils.
The study reports a SEM-based morphological characterization of the MoS2 samples (size, thichness) and of the bacteria, Zeta potential measurements of samples and bacteria suspension (in aqueous solution, presumably) and measurements of survival percentages of the bacteria cultures (in pepton water) after inoculation of MoS2.
The data indicate significant anti-bacterial activity of the material, both in form of bulk and exfoliated particulate, which would underline the relevance of MoS2 as toxic agent toward beneficial soil bacteria.
To my opinion, the work is scientifically sound and sufficiently well written. However, it is also not immune from major critics:
1.
Two-dimensional MoS2 is not a "common" material: the few-layer structure has to be obtained through some processing of the bulk material. I also have no knowledge of chemical processes that require significant amounts of MoS2 produced in actual 2D or "nanosheet-like" (few layers) form.
Hence, if the work is motivated by the idea of evidencing 2D-MoS2 as detrimental agent for soil health, the idea seems to be questionable given the actual amount MoS2 that might be involved in real cases.
2.
The actual MoS2 morphology (bulk vs nanosheet) appears to do not modify significantly the bactericide activity. In the case of B. Cereus the error bars obtained for the study of the 16 mg/mL bulk MoS2 solution indicate vales which, within one error bar, is compatible with those measured for the same bacterial culture inoculated with MoS2 nanosheets. In other words, most of the bactericide activity is due to the material itself and not to the sample morphology, even if the morpgology difference for the P. Aeruginosa tests is significant.
In consideration of the obvious fact that the few-layer samples have a much larger surface area (due to the reduction in both the mparticle tickness and lateral sizes), one can hardly infer a correlation between the thickness/low-dimensionality and the bactericide properties.
3.
The actual cause (or causes) of the bactericide activity is (are) not very clear. Several mechanisms are invoked, but in view of the small differences (bulk vs. nanosheet) of the measured data, the overall discussion seems to be confusing.
My overall evaluation is that the work is not bad in itself, but has not the depth and the completeness that I would expect for a nanomaterials paper.
Hence my recommendation is to reject the work and recomend re-submission of a new work, where the authors might take into account the following considerations and include the following analyses:
A) Include sound and well-documented support to the idea that MoS2 might represent a significant soil polluttant. While I am pretty confident that 2D-MoS2 cannot be considered a significant soil polluttant for the above-discussed reasons, the situation might be different for bulk MoS2.
In this regard the authors might discuss the industrial use of MoS2 (in the bulk form, likely!) and the possibility that this material is dispersed in waters and/or soil. In passing, they might also discuss the antibacterial properties of 2D materials and relate them to topics different than soil pollution.
B) Given that the author are putting great attention on the two-dimensional nature of MoS2 nanosheet and considering that it is very difficult to infer a correlation between the thickness/low-dimensionality and the bactericide properties from the present paper, a new work might investigate such a correletion by separating samples of nanosheets of different average thickness (for example, by repeating centrifugation steps or by employing different sonication durations) and repeat the experiment for different nanosheets (e.g. sheets with 1-5 layers, sheets with 6-10 layers, samples with more than 10 layers, bulk. This is just an example).
C) Improve the discussion on the bactericide mechanisms.
Reviewer 2 Report
The manuscript is, in principle, interesting and regards the the potential toxicity of MoS2 nanosheets. The main points to be clarified before the publication are:
1) the authors claim the presence of nanosheets but all the images reveals the presence of shapes and size out of the nanometric range, both for what concerns the lateral dimension and the thickness;
2) There is not any experimental evidence that the microplatelets evidenced by AFM and SEM are made of MoS2;
3) the thickness that can be deduced by AFM seems to be very different from those one evidenced by SEM analysis.
In such a framework, my recommendation is for a major revision that would have to address the previous points of weakness.
Reviewer 3 Report
The idea of the research fits very well with modern pro-ecological trends concerning environmental protection. I especially appreciate the concept of the project to check what happens with a great number of nanomaterials present in many products, when they reach the environment during production cycle or after use. However, I have some comments, which need to be addressed before the text will be accepted for publication in Nanomaterials.
Major comments:
- Introduction: “SiO2, TiO2, ZnO, Cu, and CuO nanoparticles were revealed as damaging various type of cells including human cells and bacteria via oxidative stress” – Cytotoxic activity of nanomaterials as you know is closely correlated to the concentration used, intensity of jones released from and many others properties. Try to refill the sentence to be more precise. Otherwise, the only conclusion from will be, that we should not use nanomaterials at all.
- Materials 2.2.:
- The description “Two soil bacterial strains” suggests soil isolates, meanwhile you used reference strains. Improve the description here and in whole text.
- “transferring loop (10 μL) of culture from tryptic soy agar” – From solid media (TSA) you can transfer only bacterial biomass, so indication the volume of loop (when you take liquid material) is unfounded. Improve.
- “Up to third transfer … A second and third transfer of the culture was washed…” – What does it mean? Did you prepare dilutions of bacterial suspension/culture? Explain.
- “The final concentration …” – For bacterial / cell suspensions a density, not concentration is assessed. Improve.
- Materials 2.3.: What volume of diluted bacterial cultures you applied to silicon pieces (10x10 mm)?
- Results and discussion (3.1.): A term “polyol phosphates” is not used in practice to describe teichoic acids structure – they are polymers of glycerol or ribitol phosphates
- Results and discussion (3.1.): “teichoic acid … to facilitate the formation of the biofilm” - This is too much of a simplification since many other polymers (e.g. polysaccharides, proteins, eDNA) and structures participate in biofilm formation (TA are only one of them). Improve.
- Results and discussion (3.3.): How many colonies you counted after bacteria exposure to MoS2 at 16 mg/ml if the reduction rate was 99,999% (it was biocidal concentration!). Is it possible, that “plateauing trends” observed results from this very high reduction rate (about 8-12 h of exposure almost all bacteria in inoculum were killed, the rest of single alive cells did not divide, so you observed plateau)
- Results and discussion (3.3.): “from the literature such as clindamycin (1.0 μg/mL), gentamicin (2.0 μg/mL), vancomycin (2.0 μg/mL) for cereus,[64] and ceftazidime (128.0 μg/mL), tobramycin (16.0 μg/mL), meropenem (16.0 μg/mL), aztreonam (128.0 μg/mL), piperacillin (64.0 μg/mL) for P.aeruginosa [65]” - There is no any information about bactericidal activity of the antibiotics in given reference. Provide the correct references.
- Check all reference numbers given in the text and provide the correct numbers!!! There is no correlation between reference in the text and reference list (e.g. Discussion: “Titley et al. [75] and Wagman et al. [76]”, while on reference list Titley et al. have no. 81 and Wagman et al. have no. 82!)
Minor comments:
- Correspondence: “…add£ressed to EMAS.” – Remove “£” and dot
- Introduction: “gfap, huC and ngn1” – use Italics for gene names
- Introduction, 3rd paragraph: Use a full name of molybdenum disulfide before chemical formula (MoS2) for the first time (the first description in the text). The Abstract in this context is treated as separate section.
- Introduction, 3rd paragraph and whole text: expand the shortcuts used in the text for the first time, e.g. hexagonal boron nitride (h-BN)
- Statistical Analysis (2.7) and the rest of the text: “survivability of Bacillus cereus and Pseudomonas aeruginosa” – whole microbial names is given only for the first time, next use shortcuts, e.g. cereus
- Figure 1: Use Italics for bacterial names
- Figure 3 and 4: Improve description of “x” axis: shortcut “t” redundant; use “h” alone (not “hr”) as in whole text
- Figure 5 and Table S1 in Supplementary Materials: Use mg/mL (instead of mg/ml) as in whole text
- Table 2: A description “± values indicate the standard deviation” remove from the title and transfer below the table
- Results and discussion: Salmonella typhimurium – it is the name of serotype (not genus), so it should be written as Salmonella Typhimurium (no Italics in second part and capital letter). Full name of genus: Salmonella enterica susp. enterica serotype typhimurium
Round 2
Reviewer 1 Report
The changes made by the authors improve the work, making it - in my opinion - good enough for publication.
Hence, i recommend to accept in the present revised version.
Author Response
Thank you for accepting the present revised version.
Reviewer 2 Report
The authors have partly responded satisfactorily, but partly have given evasive answers which reinforced my previous concerns rather than removing them. The main question that remain open is that there is not any experimental evidence that the platelets evidenced by AFM and SEM are made exclusively of MoS2.
The material used has a nominal (declared by the vendor) degree of purity of 99%. This means that 1% (at least…) is something else… Which is the yield of the used process after the separation of nanoscale materials from the supernatant layer? Which is the particle size declared by the vendor?
The increase of antibacterial activity could be related to something different from the observed shape. In principle, it would be possible that the original 1% different from Molybdenum (IV) disulfide could be all (or a significant part) of the separated nanoscale material.
Morevover, I’m indeed fully agree with this statement by referee #1: <In other words, most of the bactericide activity is due to the material itself and not to the sample morphology, even if the morpgology difference for the P. Aeruginosa tests is significant. In consideration of the obvious fact that the few-layer samples have a much larger surface area (due to the reduction in both the mparticle tickness and lateral sizes), one can hardly infer a correlation between the thickness/low-dimensionality and the bactericide properties.>
In such a context remain a fundamental issue that -as I already wrote in my first report- there is not any experimental evidence that the platelets evidenced by AFM and SEM are made exclusively of MoS2
From my point of view, it is not clear the reason for which SEM observations have not been done on the same samples (platelets) studied by AFM, with the possibility to perform easily on them also a compositional analysis by EDX.
Also structural analysis (by XRD or ED, considering the reported sizes by AFM), before and after the preparation, could help to rationalize and understand better the increase of antibacterial activity, that could be quantitative related to strain or other structural modifications introduced by the high-intensity ultrasonication process used for sample preparation.
In such a context, I continue to recommend a major revision because I cannot improve on my previous recommendation.
